Eyler *et al. Genome Biology*　　(2020) 21:174

**RESEARCH**　　　　　　　　　　　　　　　　　　　　　　　　　　　　　**Open Access**

# Single-cell lineage analysis reveals genetic and epigenetic interplay in glioblastoma drug resistance

Christine E. Eyler[1,2†], Hironori Matsunaga[2,3†], Volker Hovestadt[2,3], Samantha J. Vantine[2,3], Peter van Galen[2,3] and Bradley E. Bernstein[2,3*]

\* Correspondence: bernstein. bradley@mgh.harvard.edu
†Christine E. Eyler and Hironori Matsunaga contributed equally to this work.
²Broad Institute of Harvard and MIT, Cambridge, MA, USA
³Department of Pathology and Center for Cancer Research, Massachusetts General Hospital and Harvard Medical School, Boston, MA, USA
Full list of author information is available at the end of the article

## Abstract

**Background:** Tumors can evolve and adapt to therapeutic pressure by acquiring genetic and epigenetic alterations that may be transient or stable. A precise understanding of how such events contribute to intratumoral heterogeneity, dynamic subpopulations, and overall tumor fitness will require experimental approaches to prospectively label, track, and characterize resistant or otherwise adaptive populations at the single-cell level. In glioblastoma, poor efficacy of receptor tyrosine kinase (RTK) therapies has been alternatively ascribed to genetic heterogeneity or to epigenetic transitions that circumvent signaling blockade.

**Results:** We combine cell lineage barcoding and single-cell transcriptomics to trace the emergence of drug resistance in stem-like glioblastoma cells treated with RTK inhibitors. Whereas a broad variety of barcoded lineages adopt a Notch-dependent persister phenotype that sustains them through early drug exposure, rare subclones acquire genetic changes that enable their rapid outgrowth over time. Single-cell analyses reveal that these genetic subclones gain copy number amplifications of the insulin receptor substrate-1 and substrate-2 (IRS1 or IRS2) loci, which activate insulin and AKT signaling programs. Persister-like cells and genomic amplifications of IRS2 and other loci are evident in primary glioblastomas and may underlie the inefficacy of targeted therapies in this disease.

**Conclusions:** A method for combined lineage tracing and scRNA-seq reveals the interplay between complementary genetic and epigenetic mechanisms of resistance in a heterogeneous glioblastoma tumor model.

**Keywords:** Therapy resistance, Glioma stem cells, Epigenetic, Genetic, Tumor heterogeneity, Lineage tracing, Single-cell RNA-seq, Insulin receptor substrate/IRS

## Background

Tumors comprise heterogeneous mixtures of cells that vary in their genetic makeup and epigenetic states. Outgrowth of fit genetic subclones is a well-established mechanism of drug resistance [1] and recent work has highlighted that genetic subclones may rapidly adapt through dynamic alterations involving extrachromosomal DNA [2, 3]. Moreover, transient epigenetic changes or cell state transitions that allow tumor cells to persist through drug exposure have been described in several cancer models [4–7]. The relative contributions of these respective models and how they cooperate to confer drug resistance in cancer remains a critical question (Fig. 1a).

Genetic subclones and heterogeneous DNA methylation patterns within tumors have been characterized by sequencing-based methods [8–10]. Outgrowth of preexisting drug-resistant subpopulations has been tracked in experimental tumor models using DNA barcodes [11]. Epigenetic modes of treatment resistance have also been inferred [12, 13]. However, holistic assessment of preexisting and dynamic epigenetic and genetic drug resistance mechanisms remains an important goal that requires new methods.

Treatment resistance is of paramount concern in GBM, which inevitably relapses after treatment [14, 15]. Although these tumors frequently harbor RTK amplifications, therapies targeted against RTKs have failed in the clinic [16]. GBMs are fueled by stem-like subpopulations that tend to be relatively resistant to therapy [17] and can be modeled experimentally as gliomaspheres [18]. We therefore sought to develop strategies to evaluate the interplay between genetic and epigenetic mechanisms of RTK inhibitor resistance in gliomaspheres. We focused on a PDGFRA-amplified gliomasphere model in which a subset of cells is able to withstand PDGFR inhibition by adopting a slow-cycling persister state that is Notch-dependent and reversible [7]. We hypothesized that longer term dasatinib-resistant cell populations might depend upon both epigenetic persister phenotypes and acquired genetic events. We developed, optimized, and employed a combined single-cell RNA-seq (scRNA-seq) and lineage tracing approach to investigate.

## Results

To track outgrowth and phenotypes of clonal lineages, we combined a lentiviral transgene barcoding system [11] with single-cell RNA sequencing (scRNA-seq). A diverse library of DNA barcodes was subcloned into the 3′ UTR of a blue fluorescent protein transgene such that the barcode would be transcribed and captured in scRNA-seq data (Fig. 1b). For ease of identification, all barcode nucleotide sequences were converted to unique lineage IDs using Base32 encoding (Supplementary Fig. S1a). Deep sequencing confirmed that our starting library contained > 1 million barcodes (Supplementary Fig. S1b, see "Methods").

To maintain high barcode diversity in a patient-derived gliomasphere model (GSC8), we transduced cells with lentiviral particles bearing this library at low multiplicity of infection (Supplementary Fig. S1c). Barcode diversity was determined after targeted sequencing of genomic DNA prepared from stably transduced cells, and reads were filtered to remove putative sequencing errors (Supplementary Fig. S1d). A homogenous baseline distribution was confirmed, with each barcode accounting for < 0.11% of the total reads (Supplementary Fig. S2a, Supplementary Table S1). The starting cellular barcode diversity after lentiviral infection and puromycin selection was on average 36,

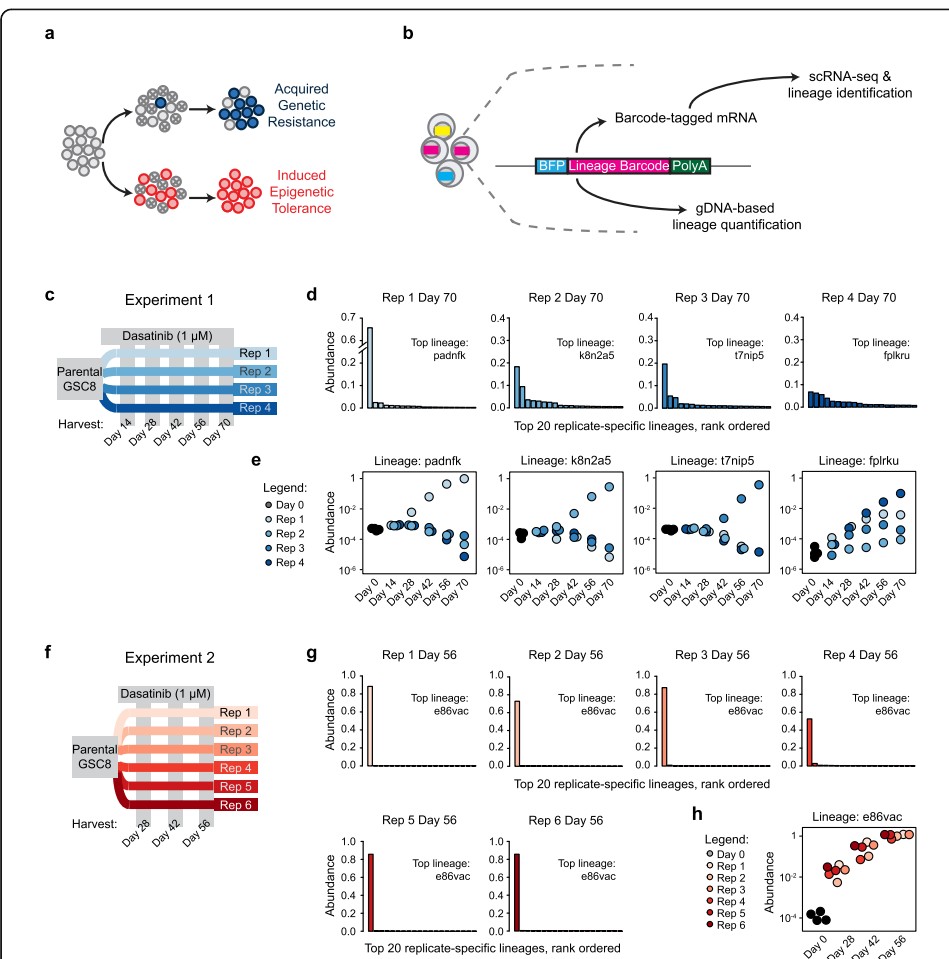

**Fig. 1** Barcode lineage tracing identifies alternative modes of drug resistance in patient-derived gliomaspheres. **a** Schematic depicts models of drug resistance involving outgrowth of fit genetic subclones or broadly induced cell state transitions that confer tolerance. **b** Schematic depicts lineage barcoding strategy wherein cells are lentivirally transduced with a unique barcode that is transcribed, enabling its identification by gDNA sequencing or scRNA-seq. **c** Experiment #1 design notates four replicates of GSC8 cells treated with the PDGFRA inhibitor dasatinib and the timepoints at which barcode abundances were analyzed by gDNA sequencing. **d** Barplots depict relative abundances of the top 20 lineages in each Experiment #1 replicate after 70 days of dasatinib treatment, per gDNA sequencing. The barcode ID of the dominant lineage in each replicate is indicated. **e** Dotplots depict relative abundances of indicated barcode lineages at successive timepoints in the different Experiment #1 replicates. Each plot shows data for a different lineage barcode that corresponds to the top lineage identified at day 70 in one of the replicates (see panel **d**). The data show that each jackpot lineage was specific to one replicate. **f–h** Similar representation as in panels **c–e** for six replicates of Experiment #2. The data indicate that a single "jackpot" lineage (e86vac) outgrew in all replicates in Experiment #2, including at earlier timepoints

580 and 84,926 lineage barcodes across replicates of each experiment, respectively (Supplementary Table S1). Thus, in comparison to prior combined lineage tracing/scRNA-seq strategies used to study organismal development [19–22], our method harnesses a highly diverse set of barcodes and is uniquely powered to detect very rare subclones in malignant cell populations [23].

GSC8 gliomaspheres harbor a PDGFRA amplification and are largely sensitive to the PDGFRA inhibitor dasatinib but a subset are able to adopt a drug persister phenotype to survive initial drug treatment [7]. To ascertain mechanisms of drug persistence and resistance, in a first experiment we distributed barcoded GSC8 cells across four

separate cultures that were treated with dasatinib continuously over a 70-day time course. Aliquots of viable cells were collected from each replicate at days 14, 28, 42, 56, and 70, and barcode distributions were assessed by targeted DNA sequencing (Experiment #1; Fig. 1c). After 70 days of dasatinib exposure, dominant clonal lineages outgrew in three of the four replicates. These dominant "jackpot" lineages made up between 18.2 and 65.5% of these replicates, representing a > 2000-fold relative expansion over the course of the experiment (Fig. 1d, Supplementary Fig. S2b; Supplementary Table S1). Their expansion was also evident in barcode distribution data for earlier time points (d42, d56) (Fig. 1e). Analysis of the fourth replicate did not reveal a jackpot lineage, but rather a spectrum of persistent lineages, all with less than 6.7% abundance at day 70 (Fig. 1d, Supplementary Fig. S2c). Notably, each of the three jackpot replicates was dominated by a different lineage. Moreover, the jackpot lineages were replicate-specific and did not show preferential fitness in any of the other replicates (< 0.012%; Fig. 1e). Control DMSO-treated populations grown from the same parental populations did not exhibit jackpot clones (Supplementary Fig. S2d). Furthermore, none of the jackpot lineages from dasatinib-treated replicates 1–3 were within the top lineages detected in DMSO-treated cells (Supplementary Fig. S2e). This implied that the exceptional fitness of the dominant lineages in dasatinib-treated cultures was conferred by distinct, replicate-specific events that occurred after splitting of the barcoded population.

We repeated this experiment by transducing a separate parental culture of GSC8 gliomaspheres with cellular lineage barcodes (Experiment #2; Fig. 1f). After confirming a homogenous baseline distribution of diverse barcodes (Supplementary Fig. S2a), we aliquoted the barcoded cells across six replicates that were exposed to dasatinib as in the prior experiment. After 56 days of drug exposure, all six replicates contained a jackpot lineage that comprised between 52.4 and 88.5% of the population (Fig. 1g, Supplementary Fig. S2f). In contrast to the prior experiment, we found that all replicates were dominated by the same lineage (lineage barcode ID e86vac). This lineage was evident as a high abundance lineage as early as the day 28 timepoint (0.4 to 2.8%, Fig. 1h). However, it was not over-represented at the starting timepoint prior to dasatinib treatment (< 0.014%). These results suggested that a single event incurred prior to splitting of the barcoded population conferred the exceptional fitness of the dominant lineage shared across these replicates.

We interpreted these lineage barcoding experiments as being representative of alternate resistance models: (1) an acquired resistance model could explain the distinct jackpot lineages evident at later timepoints in Experiment #1, (2) a preexisting resistance model could explain the single jackpot lineage shared across replicates in Experiment #2, and (3) an induced model could explain the persistence of a spectrum of lineages through the full course of Experiment #1 Replicate #4, as well as at earlier timepoints in all Experiment #1 replicates. We hypothesized that the induced resistance could reflect an epigenetic persistence mechanism, such as that we and others have described previously [6, 7, 24–26], while the instances of acquired and preexisting resistance might reflect more stable genetic alterations.

We therefore investigated the underlying alterations and mechanisms. First, we performed scRNA-seq on barcode-bearing gliomaspheres that we had collected from

Experiments #1 and #2 at day 28, reasoning that these data might reveal discriminating early features of the shared resistant subclone amongst a persister-rich background. We used the nanowell-based Seq-well technology [27] to acquire 3012 high-quality single-cell transcriptomes across both experiments at this timepoint. We were able to assign a unique lineage barcode to over 42.1% of cells with high confidence based on single-cell sequencing reads for the transcribed barcode. We identified a total of 1076 different lineage barcodes across the 1268 single cells with assigned barcodes (Fig. 2a). The most abundant lineage in the scRNA-seq data ($n = 30$, 1.9% of cells from Experiment #2) harbored the same barcode ID (e86vac) as the jackpot lineages identified in the DNA sequencing analysis of the six Experiment #2 replicates (Supplementary Fig. S3a). However, the highly-represented lineages from later timepoints in Experiment #1 were not over-represented at day 28, consistent with observations from targeted DNA sequencing that jackpot lineages from this experiment were not yet dominant at this early timepoint.

We first analyzed the transcriptional programs of individual cells by t-SNE visualization. This revealed strong overall concordance between the two experiments, both of which contained similar distributions of single-cell transcriptomes at day 28 (Fig. 2b). The t-SNE visualization revealed two main clusters of cells distinguishable by their expression of cell cycle genes (Fig. 2c, Supplementary Fig. S3b). Cells deriving from the predominant Experiment #2 lineage (e86vac) were enriched within the cell cycle-high cluster (Fig. 2d, e, Supplementary Fig. S3c), suggesting that they already had a proliferative advantage at early timepoints.

Since the e86vac lineage was the dominant lineage across multiple replicates, we hypothesized that it harbored a discrete genetic event such as a copy number alteration. To investigate mechanisms underlying the dasatinib-resistant phenotype of the dominant e86vac lineage, we aggregated single-cell transcriptomes for all cells matching this lineage and compared the expression profiles of e86vac cells against data aggregated from a cell-cycle matched cohort of cells from other lineages. We noticed that a number of upregulated genes in the e86vac lineage (e.g., COL4A1/2, SOX1, IRS2) reside within a single chromosomal band, chr13q34 (Supplementary Fig. S3d). Systematic evaluation of all chromosomal bands across the genome revealed that three bands on chromosome 13 (chr13q12, chr13q14, chr13q34) score most highly in e86vac lineage cells (Fig. 2f). We hypothesized that these transcriptional patterns reflect genetic copy number alterations, which are frequent in glioblastoma and other tumors [28, 29]. In particular, a potential role for focal chr13q34 amplification in RTK inhibitor resistance was of particular interest as recurrent amplifications of this locus have been reported in hepatobiliary, colorectal, breast, and rhabdomyosarcoma cancers [30–34]. We therefore sought to confirm a genetic alteration at this locus and its relationship to RTK inhibitor resistance.

To this end, we isolated clones from a variety of lineages identified in the dasatinib time course experiments. We flow sorted single cells from the final timepoints of both experiments into wells and expanded > 300 single-cell-derived cultures (Fig. 3a). Targeted sequencing of 75 of these cultures enabled us to assign unambiguous lineage barcodes (38 unique barcodes). These included the e86var lineage from Experiment #2 (24 clones), two jackpot lineages from Experiment #1 (padnfk and t7nip5; 6 and 2 clones, respectively), and 35 non-jackpot lineages from both experiments (43 clones). We next

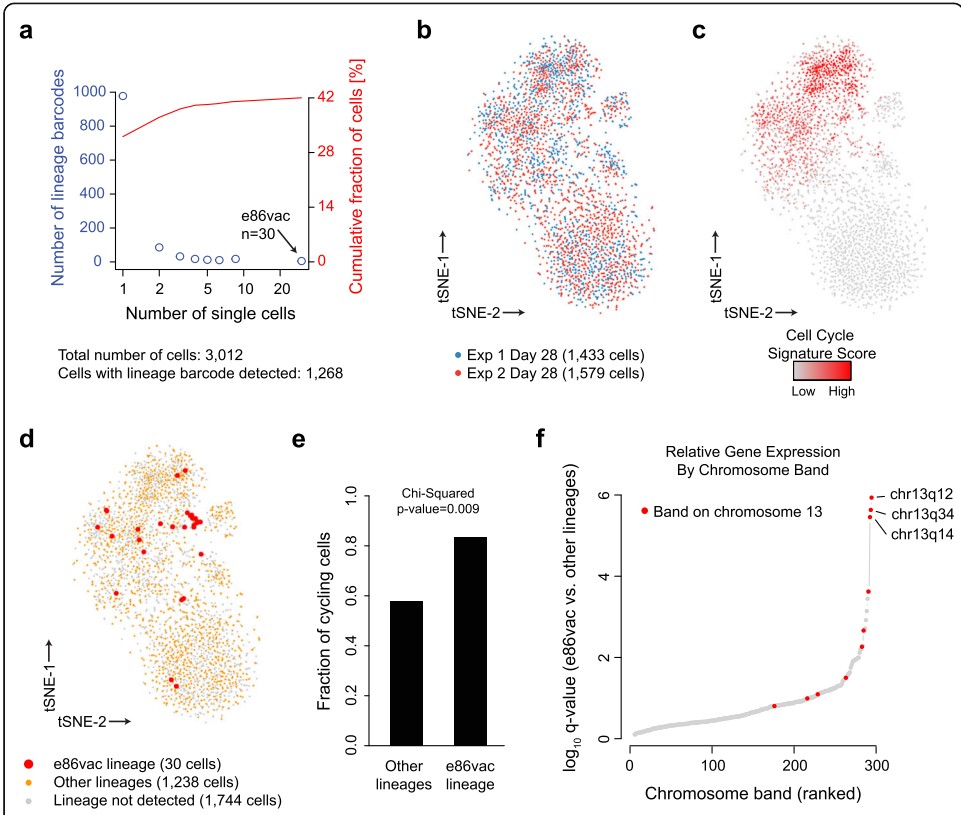

**Fig. 2** Single-cell analyses identify chromosomal amplifications in "jackpot" drug-resistant lineages. **a** Plot depicts barcode distributions in scRNA-seq data for 1268 individual cells from Experiments #1 and #2 at day 28. Blue circles indicate the number of barcodes (left axis) that were detected in the indicated number of cells (x-axis). The "jackpot" lineage identified by gDNA analysis of Experiment #2 (e86var) was detected in the largest number of cells (*n* = 30). The red line indicates the cumulative fraction of cells (right axis) for which a lineage barcode was detected. **b** t-SNE plot displays single cells (points) clustered based on their transcriptomes and colored by experiment. **c** t-SNE plot as in panel **b** with cells colored by their expression score for cell cycle signature genes. **d** t-SNE plot as in panel **b** with cells that correspond to the jackpot lineage e86vac (red) or other lineages (orange) indicated. Gray points indicate cells for which lineage could not be determined. **e** Barplot depicts fraction of cells positive for the cell cycle signature (y-axis). Data are shown for cells assigned to the jackpot e86vac or other lineages. **f** Rank-ordered plot shows chromosomal bands that harbor genes that are higher expressed in the e86vac lineage relative to all others. Chromosome 13 bands are highlighted in red and the three most significant bands are labeled

carried out low-coverage whole-genome sequencing (WGS) of 10 single-cell-derived jackpot lineages as well as 12 non-jackpot barcodes (Fig. 3b, Supplementary Table S2). Systematic analysis of copy number alterations across all 22 clones relative to parental GSC8 gliomaspheres revealed two prominent amplifications specific to the three jackpot lineages (Supplementary Fig. S4a-b). In line with our hypothesis, all e86var clones harbored a high-level amplification of chr13q34 (Fig. 3b; Supplementary Fig. S4c). Interestingly, the two jackpot clones isolated from Experiment #1 (padnfk and t7nip5) did not exhibit any copy number alterations at chr13q34, but instead contained distinct, though overlapping, amplifications at chr2q36 (Fig. 3b; Supplementary Fig. S4b,c). In contrast, we did not detect any focal high-level amplifications in the non-jackpot clones, which were nonetheless persistent and viable after 56–70 days of dasatinib exposure (Fig. 3b). All clones maintained critical copy number alterations seen in the parental cells (e.g., PDGFRA, NMYC, MDM2; Supplementary Fig. S4a,c).

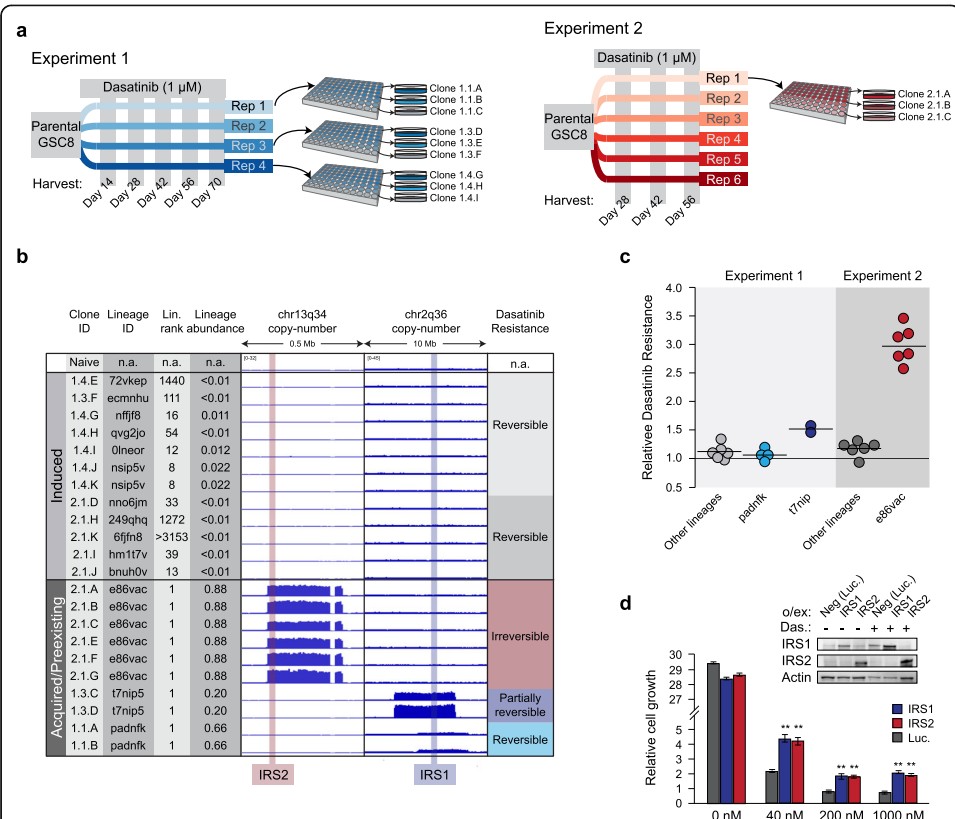

**Fig. 3** Analysis of clonal cultures identifies distinct but functionally parallel chromosomal amplifications in "jackpot" drug resistant lineages. **a** Schematic depicts isolation of single cells and derivation of clonal cultures from multiple dasatinib-treated replicates of Experiments #1 (day 70; left) and 2 (day 56; right). **b** Chart details characteristics of those clonal cultures isolated from dasatinib experiments that were analyzed by low-coverage whole-genome sequencing (WGS), compared to drug-naïve GSCs (top row). Clones that exhibited features consistent with induced drug persistence (epigenetic) in these experiments are shown in the top part of chart. Clones that exhibited features consistent with genetic (acquired or preexisting) resistance are shown in the bottom part of chart. For each single-cell-derived clone, the chart indicates the experiment and replicate from which it was derived (Clone ID), the DNA barcode identity (Lineage ID), and the rank and abundance of the respective lineage in the gDNA analysis of the corresponding experiment (day 70 for Experiment #1; day 56 for Experiment #2). Genomic tracks depict read density in low-coverage WGS over chromosomal bands 13q34 (containing IRS2) and 2q36 (containing IRS1). The final column summarizes the extent to which the clone loses dasatinib resistance after a drug-free washout period. **c** Single-cell-derived clones were subjected to a drug-free washout period and then re-challenged with dasatinib. Plot depicts growth in the presence of dasatinib, relative to drug-naïve GSC8 cells (*Y*-axis); all conditions were treated independently in at least 4 replicates. Circles represent the mean relative growth of individual clonal cultures. Horizontal black bars reflect the mean growth of clones from the indicated lineage (top). **d** Barplots depict relative growth of drug-naïve GSC8 cells transduced with IRS1, IRS2, or luciferase expression constructs (**, $p < 0.01$ by two-tailed Student's *t* test; standard error bars depicted). Cells were grown at the indicated dasatinib concentrations. Western blot shows IRS1, IRS2, and Actin protein expression in the indicated GSC8 cultures. Overexpression of IRS1 or IRS2 confers dasatinib resistance

These results suggest that, in addition to inducing a known epigenetic persister intermediate population [7], dasatinib treatment of PDGFRA-amplified GSCs can prompt outgrowth of subclonal populations with focal amplifications of chr13q34 or chr2q36. Together, these varied mechanisms of treatment response suggest that cell populations from the same patient-derived gliomaspheres may adapt to targeted RTK therapy via multiple genetic and epigenetic mechanisms. We reasoned that the chr13q34 amplification evident in e86var likely represented a relatively stable event as it was present

across all six replicates in Experiment #2. Indeed, we found that despite enduring dasatinib-induced inhibition of PDGFRA phosphorylation (Supplementary Fig. S4d), e86var clonal isolates cultured in the absence of dasatinib for > 4 weeks retained their drug-resistant phenotype when re-exposed to dasatinib (Fig. 3c). The chr2q36 amplified clones that arose differentially in Experiment #1 replicates were more variable and displayed some degree of drug resistance reversibility: clonal isolates with high copy number amplifications retained more stable dasatinib resistance than isolates with low copy number (Fig. 3c). In contrast, non-jackpot clones lost their drug tolerant phenotype entirely when cultured in the absence of dasatinib, consistent with a reversible epigenetic resistance mechanism.

To explore the mechanism by which GSC8 gliomaspheres acquire dasatinib resistance, we further investigated genes from chromosomal band chr13q34 that were upregulated in the e86vac jackpot lineage (Supplementary Fig. S3d). One of these genes was insulin receptor substrate 2 (IRS2; Fig. 3b), which has previously been identified as a low-frequency amplified gene in GBM [35] and is described as a putative driver oncogene in several other cancers [30, 32, 36–39]. Consistently, drug-naïve GSC8 gliomaspheres in which IRS2 was overexpressed exhibited robust dasatinib resistance (Fig. 3d). When we examined copy number data from the Cancer Genome Atlas [29, 35], we found that that the chr13q34 locus including IRS2 was amplified in a subset of primary glioblastomas as well as multiple other primary tumor types (Fig. 4a). Kaplan-Meier survival analysis of samples from the IDH1-wildtype, proneural subtype of GBM [29, 40, 41], which most closely reflects the GSC8 model used here [41, 42], indicated that IRS2 overexpression was associated with poor patient prognosis (Fig. 4b). We observed this correlation within the entire subtype-filtered cohort (very few patients are annotated as having received RTK inhibitors), suggesting that IRS2 may serve an oncogenic role even in the absence of targeted treatments.

Remarkably, the chr2q36 amplifications observed in the jackpot lineages padnfk and t7nip5 from Experiment #1 contained the IRS2 paralogue insulin receptor substrate 1 (IRS1; Fig. 3b, Supplementary Fig. S4b,c) [43]. IRS1 has also been implicated in cancer initiation and progression [39, 44]. We therefore hypothesized that these paralogous members of the IRS protein family both confer resistance to RTK inhibitors. Indeed, overexpression of IRS1 in our GSC model also conferred dasatinib resistance (Fig. 3d). Combined with the observation that chr13q34 (containing IRS2) and chr2q36 (containing IRS1) were the two clear dominant high-level amplifications seen in jackpot clones relative to parental cells (Supplementary Fig. S4b), these results strongly suggest that amplification of the paralogous IRS1/2 genes provide outgrowth advantage for GSC8 cells under dasatinib treatment.

Finally, we sought to characterize the mechanisms that underlie dasatinib resistance in glioblastoma. We previously documented a critical role for Notch signaling in sustaining a dasatinib-tolerant slow-cycling persister state [7]. Consistently, scRNA-seq data for the dasatinib-treated gliomaspheres revealed subsets of cells that expressed a Notch gene signature (Fig. 5a) and a persister-like gene expression signature (Supplementary Fig. 5a). Notably, these clusters were depleted of cells from the IRS2-amplified lineage e86vac, which had low Notch signature gene expression and low persister signature gene expression (Fig. 5a, b, Supplementary Fig. S5a). Further, whereas dasatinib-treated GSC8 cultures and non-jackpot lineages expressed high levels of Notch

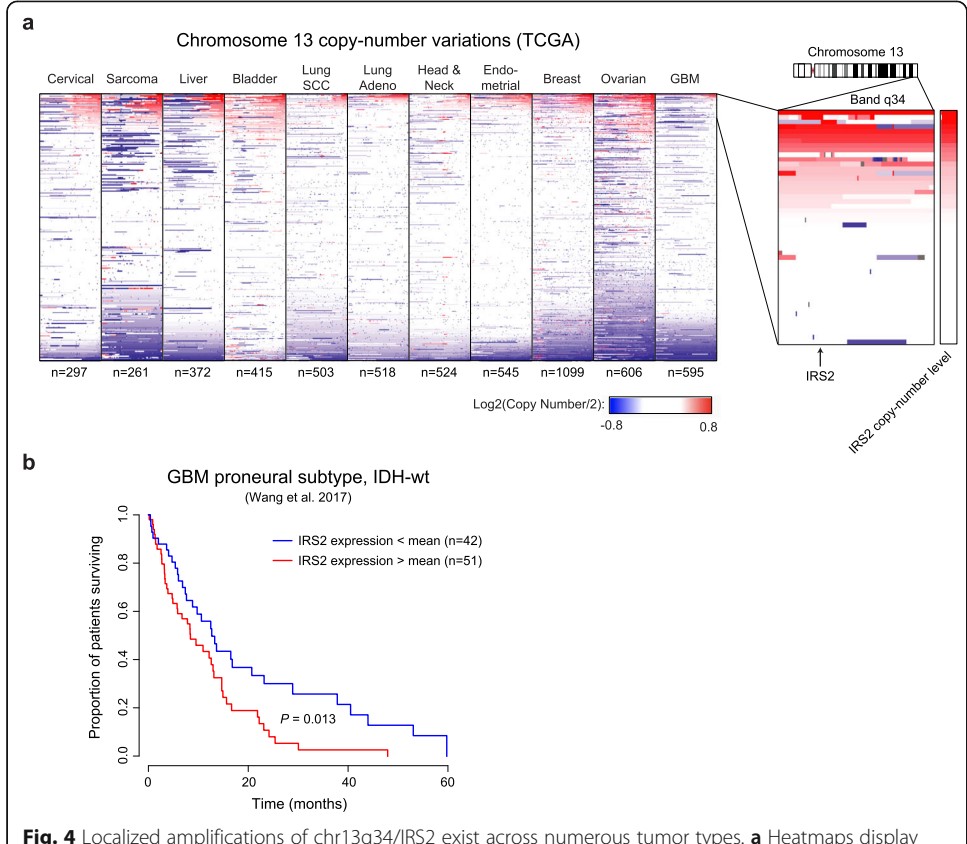

**Fig. 4** Localized amplifications of chr13q34/IRS2 exist across numerous tumor types. **a** Heatmaps display copy number alterations across chromosome 13 for indicated tumor types (blue = copy number loss, red = copy number gain). Inset shows molecular features for the subset of 50 GBMs with the highest IRS2 copy number level. **b** Kaplan-Meier survival analysis of proneural subtype glioblastomas from the TCGA project shows an inverse correlation between IRS2 expression and survival ($p = 0.013$).

intracellular domain (ICD) and downstream transcription factor RBPJK, the IRS2-amplified lineage demonstrated decreased evidence of active Notch signaling (Fig. 5c). Consistently, persister-rich GSC8 populations were sensitive to gamma secretase inhibition of the Notch pathway, while the IRS2-amplified clones were largely insensitive to Notch pathway inhibition (Fig. 5d). Thus, IRS2-mediated dasatinib resistance is independent of Notch.

IRS1 and IRS2 are both highly regulated adaptor proteins that link insulin receptor (IR) or insulin-like growth factor (IGF-1R) signaling to downstream effector pathways such as AKT and ERK (schematic in Supplementary Fig. S5b) [45]. Further analysis of the scRNA-seq data from Experiment #2 revealed an increased AKT expression signature within the IRS2-amplified lineage, relative to all other lineages (Fig. 5e,f). Western blots confirmed that in dasatinib-containing conditions, IRS2-amplified clonal lines demonstrated increased AKT activation (p-AKT) relative to parental cells and control clonal lines (Fig. 5g,h). In contrast, ERK appeared largely inactive in dasatinib-treated cells and its activation was uncorrelated with IRS2 status (Supplementary Fig. S5c). Further, IRS2-amplified clones lost sensitivity to MEK/ERK pathway inhibition (Supplementary Fig. S5d). Nonetheless, the amplified clones were highly sensitive to inhibitors of IGF-1R/IR signaling (Supplementary Fig. S5e), supporting a prominent role for IGF-1R/IR/IRS/AKT signaling in their ability to rapidly proliferate in the absence of PDGFRA signaling.

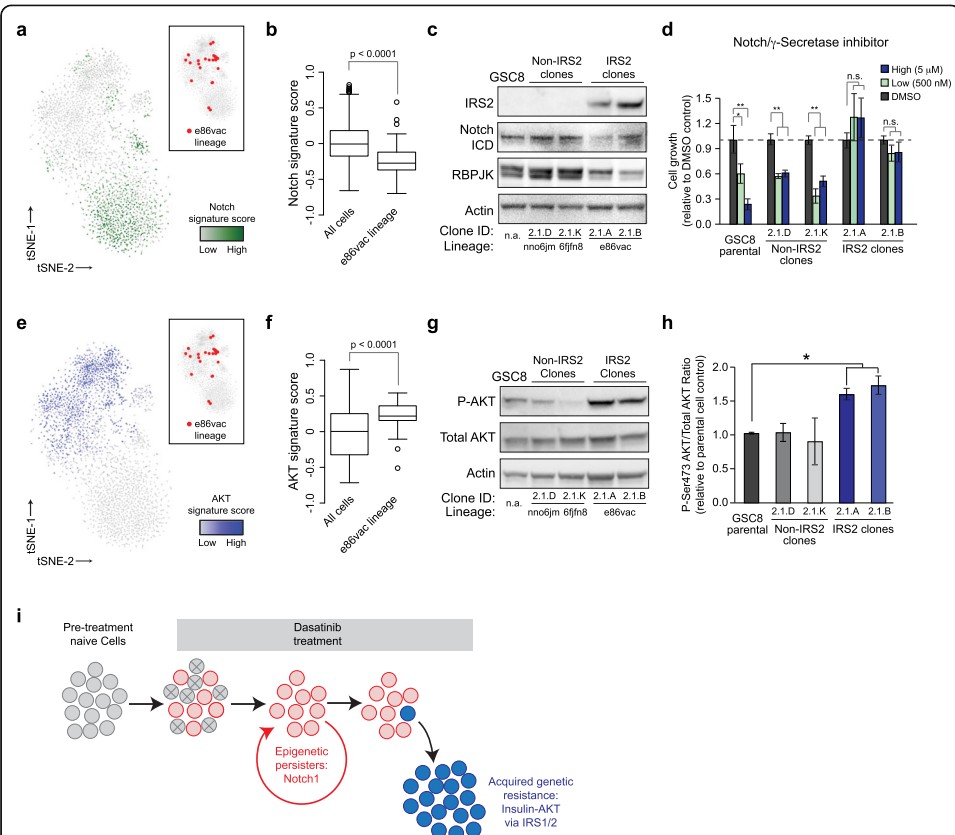

**Fig. 5** Distinct signaling programs mediate dasatinib resistance and persistence. **a** t-SNE plot displays single cells from day 28 cultures as in Fig. 2b, colored by their expression score for a Notch signaling gene signature. Inset shows the same plot with cells from the e86vac jackpot lineage highlighted in red. **b** Boxplots depict the distribution of Notch signature scores across all single cells ($n = 2982$) or e86vac lineage cells ($n = 30$). Horizontal line depicts sample median, box delimits the interquartile range, open circles indicate suspected outliers, and whiskers delimit the non-outlier distribution of data (distribution of data falling within range of median ± 1.5(interquartile range)).**c** Western blot shows expression of IRS2, activated intracellular Notch (Notch ICD), the Notch-associated transcriptional factor RBPJK and Actin. Input samples correspond to drug-naïve GSC8, non-jackpot clones from Experiment #2 and jackpot clones with IRS2 amplification grown in the presence of dasatinib. The non-jackpot "persister" lineages have high Notch ICD and RBPJK levels, while the jackpot lineage that outgrew Experiment #2 has high IRS2 expression. **d** Barplots compare growth of parental GSC8, non-jackpot clones from Experiment #2, and jackpot clones from Experiment #2 with IRS2 amplification after at least 30 days of culture in the presence of dasatinib. Dasatinib-containing cultures were treated with the indicated concentrations of γ-secretase/Notch inhibitor or DMSO control (**, $p < 0.01$; *, $p < 0.05$ by two-tailed Student's $t$ test; error bars depict the standard error of at least 4 independently treated replicates). The IRS2-amplified lineage is not dependent on Notch signaling. **e** t-SNE plot displays single cells from day 28 cultures as in Fig. 2b, colored by their expression score for an AKT gene signature. **f** Boxplots depict the distribution of AKT signature scores across single cells ($n = 2982$) or e86vac lineage cells ($n = 30$). Boxplot parameters are as in panel **a**. **g** Western blot shows levels of AKT, its active phospho-serine 463 isoform (P-AKT), and tubulin loading control. Input samples are as in panel **c**. **h** Barplot with standard error bars depicts densitometric analysis (ImageJ, NIH) of multiple ($n = 3$) separate Western blot experiments assessing AKT phosphorylation (Phospho-Serine 463) relative to total AKT in dasatinib-exposed cultures. *, $p < 0.05$ by two-tailed Student's $t$ test. **i** Schematic depicts epigenetic and genetic events proposed to confer drug tolerance, resistance, and relapse in glioblastoma. Gray circles represent drug-naïve glioma cells; red circles represent Notch-dependent persisters that tolerate therapy but proliferate slowly; blue circles represent a subclone that has acquired a genetic amplification of the IRS1 or IRS2 locus that drives AKT signaling despite upstream therapeutic inhibition

## Discussion

Here, we combined a high complexity lineage barcoding system with scRNA-seq to investigate the mechanisms by which PDGFRA-dependent stem-like GBM cells evolve resistance to PDGFRA inhibition. The data reveal alternate resistance mechanisms involving genetic or epigenetic alterations (Fig. 5i). A stable IRS2 locus amplification that enables rapid growth in the presence of dasatinib may preexist in patient-derived gliomaspheres and has been documented in primary GBMs and other tumors prior to treatment [35] (Fig. 4a). More variable copy number alterations of the IRS1 locus also facilitate rapid growth and may arise de novo during drug exposure. Finally, a rapidly reversible persister state sustains viability of a subset of cells by epigenetic mechanisms. Notch-positive persister-like cells are already present in primary tumors [7] and may bridge GBM cells through early treatment until fully drug-resistant genetic alterations arise.

These results are an important foundation for testing the hypothesis that epigenetic persistence and genetic amplifications may jointly contribute to the development of resistance to RTK inhibitors or other targeted therapies. Epigenetic persisters have been characterized as a flexible, reversible phase of therapy response [4–7]. Here, we further note that genetically distinct subclones bearing IRS1 amplifications also exhibit a degree of reversibility to dasatinib resistance that may depend on copy number and yet undetermined factors. Thus, certain genetic alterations may provide another layer of dynamic adaptability in cancers, as noted in recent work profiling rapid adaptation via extrachromosomal DNA [2, 3, 46]. Further characterization of the multiple layers of epigenetic and genetic flexibility in models of treatment response could aid in the rational design of therapeutic approaches that address both the reversible and irreversible phases of therapeutic response.

Evaluation of clinical samples from completed and ongoing clinical trials [47–49] should be prioritized to determine the contribution of IRS1/2 expression to therapy responsiveness. Additionally, prospective evaluation of the contribution of IRS1/2 overexpression to patient outcomes and tumor aggressiveness in the absence of RTK treatment could further elucidate the mechanistic specificity of these amplifications in patient tumors.

## Conclusions

Combining high complexity barcoding with single-cell RNA-seq permitted the identification of IRS2 and IRS1 copy number amplifications as novel genetic mediators of dasatinib resistance in a PDGFRA-amplified gliomasphere model, in addition to transcriptionally characterizing an intermediate Notch-high epigenetic persister population. Our study provides important insights into how these alternate modes of drug tolerance and drug resistance can conspire to overcome therapy and cause the inevitable relapse of this disease.

## Methods

### Cell models and culture

Surgical GBM specimen-derived neurosphere cultures (GBM8 a.k.a. GSC8) were isolated and previously characterized [50]. Based on gene expression and mutational analysis this model most closely resembles an IDH1-wildtype, PDGFRA-amplified

glioblastoma of the proneural subtype [41, 42]. Cells were maintained in stem cell-permissive, serum-free, non-adherent spheroid cultures using Neurobasal medium (Gibco) supplemented with N2/B27 (Gibco), penicillin/streptomycin (Gibco), Gluta-MAX (Gibco), recombinant human EGF (20 ng/mL, R & D systems), recombinant human FGF2 (20 ng/mL, R&D systems), and heparin (2000 ng/mL, Sigma). Cell dissociation was accomplished with trituration or Accutase (BD Bioscience) treatment.

### Barcode plasmid library construction

Plasmid pBA439 was a gift from Jonathan Weissman (Addgene plasmid # 85967) [51] and was modified to remove the mU6 promoter by NotI and HpaI restriction digest and self-ligation after Klenow (NEB M0210S) treatment. The resultant pBA439HN modified plasmid was used as the backbone for generating the barcode library.

To generate the semi-random barcoded insert, synthetic oligos oligoF (ATGCCG TCTCCCTAGGACTGACTGCAGTCTGAGTCTGACA GWSWSWSWSWSWSWSWSWSWSWSWSWSWSWSAGCTACGCACTCTATGC-TAGTGCTAG; W = adenine or thymine and S = guanine or cytosine) and oligoR (gcatcgtctcGAATTCCTAGCACTAGCATAGAGTGCGTAGCT) were purchased from IDT. This WSx15 semi-random sequence is referred to as the "lineage barcode." OligoF and oligoR were annealed and the double-stranded oligonucleotides were generated by extension reaction with Phusion High-Fidelity DNA polymerase (NEB). Compatible ends were generated by treating the pooled duplex barcoded insert and the vector with AvrII and EcoR1. The pooled library was then subcloned into restriction-digested vector pBA439HN. Restriction enzyme-treated pBA439HN and the barcode double-stranded oligos were mixed at a ratio 1:4 vector:insert with ~ $4 \times 10^{11}$ digested vector molecules, ligated with T4 DNA ligase (NEB) at 16 C and ligated products were purified by DNA clean & concentrator 5 kit (Zymo research). About 25% of the purified ligated products were transformed into ElectroMAX Stbl4 cells (Thermo Fisher Scientific), and cells were plated on ampicillin/LB agar plates. All plated colonies were pooled and collected, and the resultant plasmid library was purified with HiSpeed Plasmid Maxi Kit (Qiagen). The purified plasmid library was sequenced and reads filtered as described below showing a diversity of 1.056 million barcodes with 2 or greater reads/barcode.

### Lentiviral-mediated stable lineage barcoding of GBM cells

For stable incorporation of the barcoded lineage construct, pooled pBA439 library plasmid was cotransfected with psPAX2 (Gift of Didier Trono, Addgene plasmid # 12260) and VSV. G plasmids (Gift of Tannishtha Reya, Addgene plasmid # 14888) at a ratio of 3: 2:1 into 293T cells using FuGENE HD (Promega) per manufacturer's instructions. Media was changed after 6 h, and cells were cultured for 3 days. Then viral supernatant was collected and concentrated using Lenti-X concentrator (Clontech) following the manufacturer's instructions, and an estimated MOI for the viral preparation was determined with serial dilution treatment of GSCs and evaluation for tagBFP expression by flow cytometry. For each replicate, approximately $1 \times 10^6$ GSCs plated in stem cell-permissive monolayer culture with laminin (5 μg/mL Engelbreth-Holm-Swarm laminin, Sigma) were infected

with concentrated virus at an MOI of 10% for 24 h prior to selection of barcode-incorporated GSCs using puromycin (0.5 μg/mL, Life Technologies) for 72 h.

### Dasatinib treatment and passaging

Equivalent numbers ($1 \times 10^6$) of stably transduced barcoded cells were plated in replicates and treated with 1 μM dasatinib. Dissociation and passage of cells with renewal of compound-containing cell culture medium was performed at least weekly. Each plate was passed and treated independently without cross-contamination of replicates.

### Isolation, amplification, and sequencing of genome-derived barcodes

Following previously reported protocols [11], dasatinib-treated cells bearing the lineage barcode library were harvested and counted at indicated experimental time points. Genomic DNA (gDNA) was prepared with a QIAmp DNA blood mini Kit (Qiagen). The lineage barcode was amplified for multiplexed next-generation sequencing (NGS) using primers bearing Illumina adaptor and index sequences. PCR amplification of barcode was performed on half of the gDNA isolated for each replicate and time point. PCR amplification was performed in one or more reactions to ensure the equal amplification of barcodes; up to 2 μg of genomic DNA was used as a template for each PCR reaction with Titanium Taq DNA polymerase (Clontech) per manufacturer's instructions. Amplified DNA was pooled if multiple PCRs were performed for a given sample, and samples were purified with QIAquick PCR purification columns (Qiagen). Tapestation HS D5000 (Agilent) analysis was used to verify the purity and size of the indexed PCR product and quantification with Qubit dsDNA HS (Thermo Fisher) was performed prior to multiplexed NGS Illumina sequencing using a NextSeq500 instrument (Illumina). Read 1 was sequenced 38 bp (or longer) with a custom spike-in primer (CACTGACTGCAGTCTGAGTCTGACAG) with 8 bp i7 sequencing read. PhiX was added as about 10% of the total reads. Between 2.8 and 10.2 million reads were generated for each library (Supplementary Table S1). Amplification and sequencing of the barcode plasmid library was performed in a similar way.

### Computational analysis of genome-derived barcodes

Fastq files from barcode sequencing libraries were assessed for high-quality barcode reads by filtering to include only those reads that met the expected WSx15 pattern (see "Barcode Plasmid Library Construction" for details on semi-random barcode design). This sequence pattern was identified in at least 86.4% of reads in all samples. Most reads containing sequencing errors (including single base insertions or deletions) are filtered during this step as they do not match the expected pattern. This initial filtering step was performed using the grep unix command. Further analysis on summarized barcode sequences was performed using the R programming language [52]. We employed a computationally efficient method to filter out additional sequences that likely contained sequencing errors. For each sequence, we identified all 30 possible sequences with a single base mismatch (i.e., a Hamming distance of 1). If one of these 30 sequences was observed more frequently than the original sequence, the original sequence was filtered out as it likely reflects a sequencing error. Notably, this approach does not only remove single base sequencing errors, but also multiple base sequencing errors, which are in turn less frequent than single base errors (illustrated in Supplementary Fig. 1d). After filtering,

between 2,766 and 88,469 unique barcodes were detected for different samples (≥ 10 sequencing reads), greatly varying in abundance (top lineage abundance between 0.1 and 88.5%; detailed information on each sample is provided in Supplementary Table S1). Estimation of the underlying barcode complexity in the plasmid library was performed in a similar way, requiring at least two sequencing reads per barcode.

### Single-cell transcriptome profiling using Seq-well

Nanowell-based single-cell transcriptome isolation, indexing, and sequencing were performed as previously described [27, 53]. Briefly, Accutase-dissociated and filtered GSCs were counted and 10,000 single cells were loaded onto a Seq-Well nanowell array of ~ 90, 000 wells with pre-loaded beads bearing oligonucleotides containing a primer-binding sequence, cell barcode identifier, UMI, and oligo-dT sequence (Chemgenes). The array was sealed with a partially permeable polycarbonate membrane (Sterlitech Custom Order), cells were lysed in the sealed wells, and mRNA were hybridized to the oligonucleotide-conjugated beads. After liberating and pooling all beads from the array, reverse transcription was performed with Maxima H-RT (Thermo) and a template switching oligo (AAGCAGTGGTATCAACGCAGAGTGAATrGrG+G, Exiqon). Exonuclease treatment with exonuclease I (NEB) was performed prior to whole transcriptome amplification (WTA) using 2x KAPA HiFi Hotstart Readymix (KAPA) and SMART PCR primer (AAGCAGTGGTATCAACGCAGAGT, IDT). AMPure SPRI bead cleanup (0.6X, Beckman Coulter) was performed and a Nextera XT tagmentation kit (Illumina) was used to prepare indexed sequencing libraries, with a P5-SMART-Hybrid primer (AATGATACGGCGACCACCGAGATCTACACGCCTGTCCGCGGAAGCAGTGGTATCAACGCAGAGT*A*C) and a barcoded N70X primer for sample identification (CAAGCAGAAGACGGCATACGAGATXXXXXXXXGTCTCGTGGGCTCGGAGATGT) from IDT. Libraries were then sequenced using a custom Read1 primer (GCCTGTCCGCGGAAGCAGTGGTATCAACGCAGAGTAC) on a NextSeq500 instrument (Illumina). The following specifications were used for paired-end sequencing: 20 bp for read 1 (containing 12 bp cell barcode information and 8 bp UMI), 8 bp for i7 index, and 64 bp for read 2 (containing part of the transcript). Fastq files for each library were generated using bcl2fastq allowing for up to one mismatch to expected library barcodes. Identification of cell barcodes, genome alignment, and quantification of gene expression levels for each single cell was performed as previously described [53] using the splice-aware aligner STAR version 2.6.0c [54] and the R programming language version 3.3.0 [52]. For downstream analyses, we retained high-quality cells with at least 2500 UMIs and normalized expression values to a total of 10,000 per cell before log-transformation (after addition of 1).

### Barcode enrichment from single-cell transcriptomes

A portion (20%) of the WTA material was utilized for PCR enrichment of lineage barcode-containing transcripts to efficiently link cell barcodes with lineage barcodes, as previously described [53]. PCR was performed using Titanium Taq Polymerase (Clontech) and the P5-SMART PCR Hybrid PCR and Biotin-R18 Oligo (/5Biosg/GTCTCGTGGGCTCGGAGATGTGTATAAGAGACAGCATAGAGTGCGTAGCT) from IDT. Cycling parameters were as follows: (1) 95 °C for 5 min; (2) 13–15 cycles of (a) 95 °C for 30 s, (b) 55 °C for 30 s, and (c) 72 °C for 30 s, (3) final extension with

72 °C for 2 min. Purified PCR products (Qiagen PCR Purification Kit) were incubated with Streptavidin C1 Dynabeads (Thermo Fisher) in 1×BW buffer (5 mM Tris-HCl (pH 7.5), 0.5 mM EDTA, and 1 M NaCl) for 15 min at room temperature, washed with 1×BW buffer 5 times, and washed with EB buffer twice. Beads were resuspended in EB and used for PCR with P5-SMART-Hybrid and barcoded N70X primers, and Titanium Taq Polymerase with the same parameters as above. The liberated PCR products were separated from the beads and purified with 0.6X AMPure SPRI purification prior to quantification with Qubit dsDNA HS (Thermo Fisher) and quality assessment with HS D5000 Tapestation (Agilent). Following paired-end sequencing, reads were processed using the cell barcodes identified in the corresponding Seq-Well library, as described previously [53].

### Computational analysis of transcriptome-derived barcodes

Similar to genome-derived barcode libraries, sequencing reads from transcriptome-derived barcode libraries were assessed for high-quality lineage barcodes. Reads were first filtered to include only those reads that started with the target sequence of the biotinylated primer (CATAGAGTGCGTAGCT), followed by the WSx15 pattern. This initial filtering step was performed using the grep unix command. Further analysis on summarized lineage barcodes (annotated by cell barcode and UMI) was performed using the R programming language version 3.3.0 [52]. We filtered sequences according to the following five criteria: (1) The combination of cell barcode, UMI, and lineage barcode was detected by at least 20 reads. (2) The lineage barcode was identified in the genome-derived libraries of any of the four replicates at day 0 in the respective experiment. (3) The cell barcode matched a high-quality cell in the corresponding Seq-Well library. (4) For a given cell barcode/UMI combination, at least 95% of reads represented a single lineage barcode. (5) Similarly, for a given cell barcode, at least 95% of reads represented a single lineage barcode. Using these criteria, 1268 of 3012 high-quality single cells (42.1%) were matched to a defined lineage barcode.

### Clustering, signature scores, and gene set enrichment analysis

All subsequent computational analyses were performed using the R programming language version 3.3.0 [52]. For t-SNE visualization of single-cell expression profiles, we first determined variably expressed genes as previously described across both experiments ($n = 427$) and centered the log-transformed expression values for each gene. We then used 1 minus the Pearson correlation coefficient as a distance measure between cells and used the Rtsne package version 0.16 [55, 56] with the following non-default parameters: pca = F, is_distance = T, theta = 0.

For deriving single-cell gene expression signature scores, we used previously reported gene signatures that exemplified cycling cells [57], Notch1 signature (Naive GSC8 N1ICD target genes) [7], persister signature (Cluster 4) [7], and AKT signaling [58]. Signature scores were calculated as described previously [53], scoring each gene relative to the 100 genes with the smallest difference in average expression level, and averaging over all genes in the signature. Since all signatures except the cycling signatures consisted of several hundreds of genes, we refined signatures by first calculating scores using all genes, and

then only retained the 50 most informative genes (identified by correlating gene expression levels with signature scores) and recalculated scores for the shortened signature.

Gene set enrichment analysis (GSEA) for chromosomal bands was performed in bulk mode using the Liger package version 1.0 [59, 60] using default parameters. For chromosomal bands with a minimal $p$ value, we ran the analysis for a second time using $10^6$ random permutations.

### Clonal culture isolation and characterization

Populations of lineage-barcoded, dasatinib-treated cells from d70 (Experiment #1) and d56 (Experiment #2) were sorted using single-cell sorting parameters on a Sony SH800 flow cytometer. Viable cells were determined with Live/Dead near-IR staining (Invitrogen) or propidium iodide exclusion and single viable cells were sorted into U-bottom Ultralow Adhesion 96-well plates (Corning). Low clonal culture outgrowth was noted on initial attempts utilizing 100% Neurobasal with growth factor supplements, so subsequent isolations utilized 50% Neurobasal with growth factor supplements and 50% filtered conditioned media from GSC8 cells cultured in Neurobasal with growth factor supplements for 36 h with substantially higher success of clonal culture outgrowth. Clonal cultures were permitted to grow in dasatinib-free conditions with periodic dissociation and replating in serially increased volumes of fresh Neurobasal with growth factor supplements until sufficient cell numbers were available to viably freeze 2 aliquots of cells and harvest gDNA (DNeasy 96 Blood and Tissue Kit, Qiagen). gDNA was quantified with Nanodrop, and lineage barcode isolation and analysis was performed on samples with sufficient high-quality gDNA for analysis (> 10 ng/uL). Barcode amplification was performed using Q5 2X High-Fidelity Mastermix (New England Biolabs) and the following primers obtained from IDT: ClonePCR_Fwd: GATGCCTGGCGTCTACTATGT and ClonePCR_Rev: CTGATCAGCGGGTTTAAACGG with the following cycling conditions: 1 cycle of 98 °C for 30 s; 35 cycles of 98 °C for 10 s then 67 °C for 30 s then 72 °C for 30 s; elongation for 2 min at 72 °C. Gel-purified PCR products (Qiagen Gel Purification Kit) were Sanger sequenced (Quintara Biology) using the ClonePCR_Rev primer above. Clones were selected for further analysis based on sequencing quality and lineage representation, and selected clonal cultures were thawed and reassessed a second time by gDNA isolation and Sanger sequencing to confirm lineage.

### Low-coverage whole-genome sequencing

Cells were snap-frozen, gDNA was isolated using DNeasy Blood and Tissue Kit (Qiagen), and then quality and quantity of gDNA was determined by Nanodrop. In total, 50 ng gDNA was used for tagmentation and sample preparation/amplification using the Nextera DNA kit (Illumina) and Nextera i7/i5 primers (Illumina) according to the manufacturer's instructions. AMPure-cleaned DNA was quantified with Qubit HS dsDNA kit (Thermo Fisher) and PCR product size and purity assessed with Tapestation D5000 (Agilent). Samples were diluted and mixed in equimolar quantity and paired-end sequenced ($2 \times 38$ bp) in multiplexed fashion using a NextSeq500 instrument (Illumina), acquiring between 10.8 and 37.5 million reads for each library (detailed information on each sample is provided in Supplementary Table S2). Fastq files were generated using bcl2fastq and aligned to the human reference genome (hg19) using bwa mem (version 0.7.17). Alignments were

coordinate sorted and putative PCR duplicates were removed using samtools. Additionally, read pairs with an insert size larger than 1 kb were also removed. Read coverage in genomic bins of 50 kb was calculated using igvtools count (version 2.5.0). Resulting coverage tracks were subsequently normalized by scaling the median coverage of chromosome 5 to 3 (chromosome 5 did not show any apparent copy number variations between all samples). Samples were visualized using the IGV browser.

### Cell growth assays

The following inhibitors were used: dasatinib (Selleck, S1021), IR/IGFR1 inhibitor GSK1904529A (Selleck S1093), Notch/Gamma secretase inhibitor Compound E (Millipore 565790), and MEK/ERK1/2 pathway inhibitor PD98059 (Selleck S1177). Accutase-dissociated gliomaspheres were counted and plated at 1000–5000 cells/well of a 96-well plate in 125 μL Neurobasal with growth factors with either DMSO or dasatinib and/or other targeted inhibitors in at least triplicate. A set of cells of each type were analyzed for total ATP content at day 0 for normalization. Cells were permitted to grow for 7 days before they were analyzed per manufacturer's protocol by Cell Titer Glo (Promega) alongside ATP standards for end point luminescence on a Synergy HTX Platereader (BioTek). At least 4 separately treated replicates were quantified for each condition.

### Resistance quantification by AUC analysis

Relative resistance to dasatinib was evaluated by comparing the drug-response survival curves of cell populations normalized to drug-naïve GSC8 cells at various concentrations of dasatinib ranging from 1.6 nM to 5 μM. Comparative AUC analysis was performed using computeAUC in the PharmacoGx package in R [52] and normalizing AUC values to that of drug-naïve GSC8 cells from the same experiment. A total of 12 clonal cultures from Experiment #1 and 12 from Experiment #2 were analyzed (including but not limited to the clones evaluated by WGS in Fig. 3b).

### Lentiviral-mediated overexpression of IRS1/2 in GBM cells

For stable incorporation of overexpression constructs, 10.5 μg pSMAL plasmid [61] with control (pSMAL-luciferase) or IRS1/2 overexpression modules were cotransfected with 1.9 μg REV, 3.8 μg RRE, 2.7 μg VSV-G, and 4.5 μg pADV plasmids per 10 cm dish with FuGENE HD (Promega) per manufacturer's instructions. Media was changed 8 h after transfection, then harvested and concentrated at 48 h post-transfection. Approximate viral stock MOI was estimated by GFP positivity by flow cytometric analysis after serial dilution treatment of GSCs. GSCs were treated with equivalent MOIs of control (pSMAL-Luciferase) and pSMAL-IRS1 and pSMAL-IRS2 virus, and stable populations with similar GFP positivity were obtained by flow sorting on a Sony SH800S Flow Sorter and IRS overexpression levels were verified by Western blotting (Fig. 3d).

### Western blot analysis

Cells were treated with DMSO or dasatinib continually for at least 30 days and then harvested for analysis. PBS-washed, live cells were snap-frozen and lysed using RIPA buffer (Thermo Fisher) supplemented with HALT protease and phosphatase inhibitors

(Thermo Fisher) on ice for 20 min, lysates were homogenized by drawing through a 23-Gauge needle, and debris was removed by centrifugation. Protein content was quantified in triplicate by BCA assay (Thermo Fisher). Equivalent lysate quantities (25–30 μg/well) were brought to the same volume, prepared with 4X LDS sample buffer (Thermo Fisher) and TCEP reducing agent (Thermo Fisher) per manufacturer's instructions, and treated at 95 °C for 5 min. Samples and molecular weight ladder (BioRad Kaleidoscope Protein Ladder) were loaded on a NuPAGE 4–12% gradient bis-tris precast polyacrylamide gel (Thermo Fisher) and running at 90–125 V until the dye front passed from the gel. Protein was transferred at 20 V for 6 min using the iBlot transfer system (Thermo Fisher) onto prepared nitrocellulose iBlot transfer stacks. After blocking for 1 h in 5% BSA in TBST at room temperature, membranes were incubated in indicated concentrations of antibodies overnight at 4 °C in TBST+ 5% BSA. Antibodies were used at following concentrations: p-Ser473 AKT (Cell Signaling Technologies, CST #4060, 1: 1000, lot 23), total AKT (CST #2920, 1:1000, lot 8), p-ERK1/2 (CST #4370, 1:1000, lot 24), total ERK (CST #9107, 1:1000, lot 10), cleaved Notch ICD (CST #4147, 1:1000, lot 6 and 7), RPBJK (Abcam ab25949, 1:2000, lot GR3240322-1), IRS2 (Abcam ab134101, 1:2000, lot GR219213-16), IRS1 (Abcam ab40777, 1:2000, lot GR278510-7). Membranes were extensively washed with TBST, incubated with secondary antibodies at 1:2500 (Goat anti Rabbit IgG Starbright Blue 700, Goat anti Mouse IgG Dylight 800, Biorad or HRP anti-Rb/Mo, CST #7074 and 7076) and 1:2500 rhodamine-conjugated FAB anti-actin or anti-tubulin (Biorad) for 1 h at room temperature, and then washed extensively with TBST. Fluorescently labels were visualized with GelDoc imager (BioRad). Membranes incubated with HRP-conjugated antibodies were treated for 1–5 min with ECL prepared according to the manufacturer's instructions (Millipore) prior to chemiluminescent imaging with GelDoc Imager (BioRad). Uncropped Western blot data, including ladders, are displayed in Supplementary Fig. S6.

### Analysis of tumor copy number and expression data from TCGA

To generate tumor type-specific heatmaps of copy number data across tumor types, the Cancer Genome Atlas (TCGA) was accessed via UCSC's xenabrowser.net. TCGA data for each tumor type was sorted by IRS2 copy number level and chromosome 13 copy number data visualized using custom display settings (origin = 0, threshold = 0.3, saturation = 0.8).

Survival analysis of TCGA GBM samples was performed using processed exon array gene expression data obtained from the NCI GDC data portal (https://portal.gdc.cancer.gov). Sample annotation was obtained from the supplementary tables of the primary publication [29, 35]. We only included IDH-wildtype cases associated with the proneural subtype as identified by previously reported subtype-specific gene lists in IDH1 WT tumors [41]. Kaplan-Meier analysis was performed using the survival package in R, splitting cases in two equally sized groups based on their expression for IRS2.

### Supplementary information

---

**Additional file 1.** Supplemental Figures and Legends (Figs. S1-S6). Contains compiled supplementary figures and legends referenced in the main text.

---

**Additional file 2.** Supplemental Table S1. Contains summary data (e.g., number of reads, number of barcodes, top barcode ID and prevalence, etc.) from each lineage barcode gDNA library sequenced and reported in the main text.

**Additional file 3.** Supplemental Table S2. Contains summary data for each low-pass WGS library sequenced and reported in the main text.

**Additional file 4.** Review history.

## Acknowledgements
We thank J.A. Verga and R. Boursiquot for technical assistance and B. Liau, C. Sievers, F. Najm, L. Gaskell, and other Bernstein lab members for helpful discussions.

## Review history
The review history is available as Additional file 4.

## Peer review information

## Authors' contributions
C.E.E., H.M., and P.V.G. conceptualized and designed the experiments. C.E.E., H.M., and S.J.V. optimized and executed the experiments. V.H. performed the computational analysis. B.E.B. provided senior guidance. C.E.E. and B.E.B. wrote the manuscript with assistance from other authors, all of whom approved the final submission.

## Authors' information
Present addresses (if different from title page):
C.E.E.: Department of Radiation Oncology, Duke University Hospital, Durham, NC, USA
H.M.: Oncology Research Laboratories II, Daiichi Sankyo Co., Ltd. Tokyo, Japan.
P.v.G.: Division of Hematology, Department of Medicine, Brigham and Women's Hospital and Harvard Medical School, Boston, MA, USA

## Funding
C.E.E. is supported by an appointed KL2 award from Harvard Catalyst | The Harvard Clinical and Translational Science Center (National Center for Advancing Translational Sciences, National Institutes of Health Award KL2 TR002542). The content is solely the responsibility of the authors and does not necessarily represent the official views of Harvard Catalyst, Harvard University and its affiliated academic healthcare centers, or the National Institutes of Health.
B.E.B. is the Bernard and Mildred Kayden Endowed MGH Research Institute Chair and an American Cancer Society Research Professor. This research was supported by the National Cancer Institute (DP1CA216873) and SCC award I9-A9-071.
None of the funding bodies were directly involved in the design of the study, nor in the collection, analysis or interpretation of the data, or writing of the manuscript.

## Availability of data and materials
Sequencing reads and processed data that supports the findings of this study have been deposited in GEO with the accession code GSE142119 [62]. To access data, go to https://www.ncbi.nlm.nih.gov/geo/query/acc.cgi?acc=GSE142119. The authors declare that data processing steps are described in detail within the methods such that analysis can be replicated.
The authors declare that all other data supporting the findings of the study are available within the paper and its supplementary information files.

## Ethics approval and consent to participate
Not applicable.

## Consent for publication
Not applicable.

## Competing interests
B.E.B. declares outside interests in Fulcrum Therapeutics, 1CellBio, HiFiBio, Arsenal Biosciences, Cell Signaling Technologies, BioMillenia, and Nohla Therapeutics. H.M. is a paid employee of Daiichi Sankyo Co. The other authors declare that they have no competing interests.

## Author details
[1]Department of Radiation Oncology, Massachusetts General Hospital and Harvard Medical School, Boston, MA, USA. [2]Broad Institute of Harvard and MIT, Cambridge, MA, USA. [3]Department of Pathology and Center for Cancer Research, Massachusetts General Hospital and Harvard Medical School, Boston, MA, USA.

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

## 