## [**Additional file 4.** Review history. · Genome Biology]

Review History

First round of review

Reviewer 1

Are you able to assess all statistics in the manuscript, including the appropriateness of statistical tests used? Yes, and I have assessed the statistics in my report.

Comments to author:

This is an excellent manuscript that uses cutting edge single-cell lineage tracing and RNA-seq technology to elegantly separate genetic and epigenetic drug resistance mechanisms in glioblastoma. In my opinion, the paper would benefit from a few minor revisions that do not require further experimentation:

- 1) There are several previous reports (including from the authors) characterizing the phenotype of human gliomas by scRNA-seq. It would be useful to place the untreated GSC8 neurospheres used in these studies in the context of what is known from this prior work. Presumably, GSC8s are somewhat proneural given the PDGFRA amplification, but how heterogeneous are they? Do they harbor subpopulations that resemble other subtypes or neural lineages? Overall, it would be helpful if the authors could comment on the extent to which GSC8s resemble human glioma.
- 2) In Fig. 3b, the authors analyze copy number alterations in Chr. 13 by low-pass WGS, which they had hypothesized to be amplified in certain clones. It would be useful to show this analysis for the rest of the genome perhaps as a supplementary figure. Were there other loci with CNVs in these clones?
- 3) In the methods section, the authors refer to a "WSx15 pattern" without explanation. It would be helpful to explain what this is. Similarly, the authors refer to their ability to filter sequencing errors in the barcode set. Presumably this is because the barcodes have a large minimum pairwise Hamming distance. What is this distance?
- 4) The authors use a book chapter from Fan (ref 54) as a reference for the Liger algorithm. I believe the correct primary reference is Welch et al, Cell, 2019 (assuming there isn't another algorithm with the same name).

Reviewer 2

Are you able to assess all statistics in the manuscript, including the appropriateness of statistical tests used? Yes, and I have assessed the statistics in my report.

Comments to author:

Eyler et al sought to delineate mechanisms of resistance to receptor tyrosine kinase inhibition in glioblastoma, in specific targeting PDGFRA with dasatanib. By combining single-cell barcoding and matched single-cell RNA sequencing, they are able to functionally annotate the most dominant clones following a period of GBM cell culturing under dasatanib treatment conditions. This is an interesting method that adds a new approach to studies of treatment resistance. There are some concerns about the validity of the conclusions drawn and interpretations of the results. The following points outline these:

Minor:

- The use of a single cell line PDGFRA amplified cell line GSC8 limits translation of findings to PDGFRA amplified patients, can the same results be observed in other PDGFRA amplified glioma lines?

Major comments

1. The absence of an untreated control limits the impact of the results. Especially in the first experiment, it is not clear whether the results would have been much different if cells would have been allowed to grow for 70days, in absence of dasatinib. At minimum, the manuscript should be edited to downplay the findings accordingly, for example with respect to labeling majority Experiment #1 clones as jackpot clones. Ideally, data on untreated cells would be included for comparison.
2. Fig 3C suggests that the 'jackpot' lineage from experiment 1.3 is not dasatinib resistant, whereas the resistance of replicate 1.1 is ~1.5x relative to the 3.0 of the lineage dominating the second experiment. This suggests that either labeling these lineages as driving resistance is not warranted, or that the cultures are not dasatinib resistant. To confirm that PDGFRA is accurately being targeted and that the post-treatment (after day #56/#70) resistance is associated with PDGFRA, please show PDGFRA DNA copy number and (phospho-)protein levels at both time points.
3. If IRS2 is associated with relatively poor survival (in a very poor survival patient group), is it specifically associated with PDGFRA resistance or a generic marker/driver of very aggressive tumors? This would seem in line with their Fig. 5 observation that IRS2 marks PDGFRA independent dasatinib resistance.
4. The methods describe that the proneural GBM group was filtered for high EGFR/PDGFRAs expressors. What was the rationale for this filter? Needs to be described/explained in results in order to interpret result appropriately.

Minor

1. "These findings suggest that cell populations from the same patient-derived gliomaspheres can adapt to targeted RTK therapy via multiple genetic and epigenetic mechanisms." This statement is not justified by data, as there is no epigenetic data (only suggestions) and the genetic mechanism appears to be not dasatinib specific.

Reviewer #1:

This is an excellent manuscript that uses cutting edge single-cell lineage tracing and RNA-seq technology to elegantly separate genetic and epigenetic drug resistance mechanisms in glioblastoma. In my opinion, the paper would benefit from a few minor revisions that do not require further experimentation:

We appreciate the positive comments as well as the helpful clarifications and analyses suggested by this reviewer, which have been addressed in the revised manuscript.

1) There are several previous reports (including from the authors) characterizing the phenotype of human gliomas by scRNA-seq. It would be useful to place the untreated GSC8 neurospheres used in these studies in the context of what is known from this prior work. Presumably, GSC8s are somewhat proneural given the PDGFRA amplification, but how heterogeneous are they? Do they harbor subpopulations that resemble other subtypes or neural lineages? Overall, it would be helpful if the authors could comment on the extent to which GSC8s resemble human glioma.

We agree with the reviewer's assumption that GSC8s likely resemble a proneural phenotype, given that amplification of PDGFRA is described as one of the main oncogenic drivers in this molecular subtype. To test this assumption in more detail we evaluated previously-described, subtype-specific gene signatures(1) within our single cell transcriptomic data. This analysis confirmed that proneural genes are most highly expressed in GSC8 cells, whereas classical and mesenchymal genes have much lower expression (Figure R1, left). Genes specific to the neural subtype show intermediate expression, however this subtype has been omitted in more recent subtype classifications(2). We further investigated the cellular heterogeneity in our single cell transcriptomic data. Unsupervised clustering analysis and heatmap visualization demonstrates that GSC8 cells are overall homogeneous in the expression of subtype-specific signature genes, and do not harbor subpopulations that resemble other subtypes (Figure R1, right). We present these cell model characteristics in the revised Methods, "Cell models and Culture".

2) In Fig. 3b, the authors analyze copy number alterations in Chr. 13 by low-pass WGS, which they had hypothesized to be amplified in certain clones. It would be useful to show this analysis for the rest of the genome perhaps as a supplementary figure. Were there other loci with CNVs in these clones?

The reviewer raises an important point. We have now significantly expanded our copy number analysis in the parental and derivative clones (revised text and new Supplementary Fig. S4). We find that the parental, dasatinib-naïve GSC8s display known high-level amplifications at the PDGFRA, MYCN and MDM2 loci, as well as several other chromosomal or sub-chromosomal copy number alterations (Supplementary Fig S4a). Low pass WGS read coverage of the clonally-derived dasatinib resistant clonal cultures was normalized to that of the parental cells. This showed that all clones retained the PDGFRA, MYCN, and MDM2 amplifications seen in the parental cells (Supplementary Fig. S4c). However, the Chr2q36/IRS1 and Chr13q34/IRS2 locus amplifications seen in lineages t7nip5/padnfk and e86vac, respectively, were the only prominent newly-arising amplifications in any of our clonal cultures (Supplementary Fig. S4b). These amplifications were specific to clones derived from dasatinib-treated cultures.

3) In the methods section, the authors refer to a "WSx15 pattern" without explanation. It would be helpful to explain what this is. Similarly, the authors refer to their ability to filter sequencing errors in the barcode set. Presumably this is because the barcodes have a large minimum pairwise Hamming distance. What is this distance?

We have included more in-depth descriptions of these two concepts within the Methods.

- 1) The WSx15 pattern refers to the semi-random lineage barcode sequence WSWWSWSWSWSWSWSWSWSWSWSWSWSWSWS where W=adenine or thymine and S=guanine or cytosine (clarified in Methods in "Barcode Plasmid Library Construction" subheading)
- 2) Most sequencing errors (including single base insertions or deletions) are filtered because they do not match the WSx15 pattern. However, single-base sequencing errors could still inappropriately give rise to valid barcode sequences. We added the following explanation to the Methods under the heading "Computational Analysis of Genome-derived Barcodes" describing our filtering strategy: "We employed a computationally efficient method to filter out additional sequences that likely contained sequencing errors. For each sequence, we identified all 30 possible sequences with a single base mismatch (i.e. a Hamming distance of 1). If one of these 30 sequences was observed more frequently than the original sequence, the original sequence was filtered out as it likely reflects a sequencing error. Notably, this approach not only removes single base sequencing errors, but also multiple base sequencing errors, which are in turn less frequent than single base errors". We included a new panel in supplementary data to illustrate this filtering step (Supplementary Fig. S1d and Figure R2).

4) The authors use a book chapter from Fan (ref 54) as a reference for the Liger algorithm. I believe the correct primary reference is Welch et al, Cell, 2019 (assuming there isn't another algorithm with the same name).

There are indeed two Liger algorithms; we refer to the Lightweight Iterative Gene EnRichment package by J. Fan. We have now included a direct reference to the source code from Fan et al(3).

Reviewer #2:

Eyler et al sought to delineate mechanisms of resistance to receptor tyrosine kinase inhibition in glioblastoma, in specific targeting PDGFRA with dasatinib. By combining single-cell barcoding and matched single-cell RNA sequencing, they are able to functionally annotate the most dominant clones following a period of GBM cell culturing under dasatinib treatment conditions. This is an interesting method that adds a new approach to studies of treatment resistance. There are some concerns about the validity of the conclusions drawn and interpretations of the results. The following points outline these:

We appreciate the reviewer's enthusiasm for our method and its potential utility in studies of treatment resistance. The revised manuscript contains new data, analyses and discussion to support our interpretation of the results and reinforce the validity of our conclusions.

Minor:

- The use of a single cell line PDGFRA amplified cell line GSC8 limits translation of findings to PDGFRA amplified patients, can the same results be observed in other PDGFRA amplified glioma lines?

We previously characterized other PDGFRA-amplified GSC lines; some have intrinsic dasatinib resistance, while others can acquire resistance (e.g., GSC87)(4). However, we have not performed detailed analysis of the underlying genetics and mechanisms in other models. An important future goal (outside the scope of the current manuscript) is to systematically compare intrinsic and acquired resistance mechanisms across tumors and models. We now address this point in our revised Discussion.

Major comments

1. The absence of an untreated control limits the impact of the results. Especially in the first experiment, it is not clear whether the results would have been much different if cells would have been allowed to grow for 70 days, in absence of dasatinib. At minimum, the manuscript should be edited to downplay the findings accordingly, for example with respect to labeling majority Experiment #1 clones as jackpot clones. Ideally, data on untreated cells would be included for comparison.

In new data and analyses for the revision, we examined barcode distributions in GSC8 cells grown in the absence of dasatinib (new Fig. S2). These control cells were derived from the same parental barcoded population as Experiment #1 and also expanded for 70 days. In contrast to the dasatinib-treated cultures, the abundance of the "jackpot lineages" (padnfk, k8n2a5 and t7nip5) actually decreased slightly relative in these control cultures, relative to initial proportions (Fig. S2f). None of the jackpot lineages were within the top 20 lineages at d70 of the DMSO control replicates and we did not observe outgrowth individual lineages above 7% in the controls (Fig. S2e). These data support the notion that the expansion of subclones in the dasatinib-containing cultures indeed reflected dasatinib resistance rather than a general fitness under conditions of prolonged cell culture

2. Fig 3C suggests that the 'jackpot' lineage from experiment 1.3 is not dasatinib resistant, whereas the resistance of replicate 1.1 is ~1.5x relative to the 3.0 of the lineage dominating the second experiment. This suggests that either labeling these lineages as driving resistance is not warranted, or that the cultures are not dasatinib resistant. To confirm that PDGFRA is accurately being targeted and

that the post-treatment (after day #56/#70) resistance is associated with PDGFRA, please show PDGFRA DNA copy number and (phospho-)protein levels at both time points.

The reviewer raises important points regarding the biology and semantics of therapy resistance. We agree that resistance of the IRS1-amplified lineages from Experiment #1 appears partially or wholly reversible, and highlight this within the Results section. This is in contrast to resistance of the IRS2-amplified lineage from Experiment #2 e86vac, which appears irreversible. The dasatinib-free outgrowth conditions used to expand the clonal cultures after single-cell sorting were such that reversibility would be magnified when resistance was evaluated in Figure 3C, but we agree that this variability is of interest in future studies and have commented on this within the Discussion.

We performed Western blotting data that confirmed that dasatinib effectively inhibited PDGFRA-phosphorylation in the IRS2-amplified clonal e86vac cultures, despite the clone's apparent irreversible resistance to dasatinib (Figure R3). Additionally, our newly-included analysis of CNVs verifies that all of the examined clonal cultures retained copy number amplification at the PDGFRA genomic locus at levels comparable to the parental GSC8 cells (Supplementary Fig. S4c). This suggests that the apparent dasatinib resistance (reversible or irreversible) did not derive from loss of amplification at the PDGFRA locus, nor from loss of dasatinib-mediated inhibition of tyrosine kinase activity.

Finally, we appreciate the detailed attention of the Reviewer to this figure, and have identified that the clonal cultures derived from the fit lineages of experiments 1.3 and 1.1 were mislabeled in the submitted Figure 3b/c. The lineage labels in this figure have been corrected.

3. If IRS2 is associated with relatively poor survival (in a very poor survival patient group), is it specifically associated with PDGFRA resistance or a generic marker/driver of very aggressive tumors? This would seem in line with their Fig. 5 observation that IRS2 marks PDGFRA independent dasatinib resistance.

We appreciate the question and now discuss potential factors contributing to the association between IRS2 expression, PDGFRA inhibition, and poor prognosis in Discussion. Our TCGA analysis suggests that IRS2 expression is associated with poor prognosis in proneural IDH1-WT tumors as a whole (Fig. 4b). However, with available TCGA data, we cannot determine whether IRS2-expression is specifically linked to PDGFRA inhibitor resistance in treated patients, and as far as we are aware these data are not available from published or ongoing trials evaluating dasatinib in PDGFRA-amplified GBM.

4. The methods describe that the proneural GBM group was filtered for high EGFR/PDGFR α expressors. What was the rationale for this filter? Needs to be described/explained in results in order to interpret result appropriately.

The reviewer's insightful comment prompted us to re-evaluate available TCGA data without filtering for high EGFR/PDGFR α expressors. We used updated GBM subtype annotations to identify IDH1-WT proneural subtype tumors(2), including more than double the cases as in our previous analysis. We now observe that IRS2 expression is correlated with poor prognosis in the entire subtype irrespective of RTK expression. We have more thoroughly clarified our TCGA subject selection and subtype stratification within the "Analysis of Tumor Copy Number and Expression Data from TCGA" section of the Methods, and now include an updated Fig. 4b with these results.

Minor

1. "These findings suggest that cell populations from the same patient-derived gliomaspheres can adapt to targeted RTK therapy via multiple genetic and epigenetic mechanisms." This statement is not justified by data, as there is no epigenetic data (only suggestions) and the genetic mechanism appears to be not dasatinib specific.

We have revised the statement to acknowledge the reviewer's concern to say the following: "These results suggest that, in addition to inducing a known epigenetic persister intermediate population(4), dasatinib treatment of PDGFR α -amplified GSCs prompts outgrowth of subclonal populations with focal amplifications of chr13q34 or chr2q36. Together, these varied mechanisms of treatment response suggest that cell populations from the same patient-derived gliomaspheres may adapt to targeted RTK therapy via multiple genetic and epigenetic mechanisms."

RESPONSE LETTER REFERENCES:

1. Verhaak RG, Hoadley KA, Purdom E, Wang V, Qi Y, Wilkerson MD, et al. Integrated genomic analysis identifies clinically relevant subtypes of glioblastoma characterized by abnormalities in PDGFR α , IDH1, EGFR, and NF1. *Cancer Cell*. 2010;17(1):98-110.
2. Wang Q, Hu B, Hu X, Kim H, Squatrito M, Scarpace L, et al. Tumor Evolution of Glioma-Intrinsic Gene Expression Subtypes Associates with Immunological Changes in the Microenvironment. *Cancer Cell*. 2017;32(1):42-56 e6.
3. Fan J. LIGER: Lightweight Iterative Geneset Enrichment. 1.1 ed2017.
4. Liao BB, Sievers C, Donohue LK, Gillespie SM, Flavahan WA, Miller TE, et al. Adaptive Chromatin Remodeling Drives Glioblastoma Stem Cell Plasticity and Drug Tolerance. *Cell Stem Cell*. 2017;20(2):233-46 e7.

Second round of review

Reviewer 1

In my opinion, the authors have done a thorough job of responding to my previous comments.

Reviewer 2

The authors have done a great job addressing the comments on the initial review with the new addition of Fig S2 and S4. I am not sure why they decided to not include the figure R3 included in the rebuttal as an addendum to Fig S4, as it shows that phospo-PDGFR α following dasatinib treatment is strongly decreased which seems valuable to interpret the results. But that is up to the authors to decide.